

# Incidence of hemoparasitic infections in cattle from central and northern Thailand

Pongpisid Koonyosying[1,2], Amarin Rittipornlertrak[2,3],
Paweena Chomjit[2], Kanokwan Sangkakam[2], Anucha Muenthaisong[2],
Boondarika Nambooppha[2,4], Wanwisa Srisawat[1,2], Nisachon Apinda[2,3],
Tawatchai Singhla[3] and Nattawooti Sthitmatee[2,4,5]

[1] Graduate School of Veterinary Science, Faculty of Veterinary Medicine, Chiang Mai University, Muang, Chiang Mai, Thailand
[2] Laboratory of Veterinary Vaccine and Biological Products, Faculty of Veterinary Medicine, Chiang Mai University, Muang, Chiang Mai, Thailand
[3] Department of Food Animal clinics, Faculty of Veterinary Medicine, Chiang Mai University, Muang, Chiang Mai, Thailand
[4] Department of Veterinary Bioscience and Veterinary Public Health, Chiang Mai University, Faculty of Veterinary Medicine, Muang, Chiang Mai, Thailand
[5] Excellence Center in Veterinary Bioscience, Chiang Mai University, Muang, Chiang Mai, Thailand

## ABSTRACT

**Background:** Hemoparasites, such as *Babesia* spp., *Theileria* spp. and *Anaplasma* spp., can negatively affect the health of farm animals resulting in significant losses in production. These losses inherently affect the economics of the livestock industry. Since increases in the severity of vector-borne diseases in the southeast Asian region have been reported, investigations of parasitic epidemiology in Thailand will be necessary to improve the existing parasite control strategies for blood parasitic infections. This study aims to investigate incidences of bovine hemoparasites throughout central and northern Thailand by focusing on areas of high-density cattle populations.

**Methods:** Blood parasitic infections among cattle were screened and identified by microscopic examination. Anemia status was then determined by evaluation of the packed cell volume (PCV) of each animal. Furthermore, blood parasites were detected and identified by genus and species-specific primers through the polymerase chain reaction method. Amplicons were subjected to DNA sequencing; thereafter, phylogenetic trees were constructed to determine the genetic diversity and relationships of the parasite in each area.

**Results:** A total of 1,066 blood samples were found to be positive for blood parasitic infections as follows: 13 (1.22%), 389 (36.50%), and 364 (34.15%) for *Babesia bovis*, *Theileria orientalis*, and *Anaplasma marginale*, respectively. Furthermore, multiple hemoparasitic infections in the cattle were detected. The hematocrit results revealed 161 hemoparasitic infected samples from 965 blood samples, all of which exhibiting indications of anemia with no significant differences. Sequence analysis of the identified isolates in this study revealed that *B. bovis rap-1*, four separate clades of *T. orientalis msps*, and *A. marginale msp4* exhibited considerable sequence similarity to homologous sequences from isolates obtained from other countries. Sequence similarity ranged between 98.57–100%, 83.96–100%, and 97.60–100% for *B. bovis rap-1*, *T. orientalis msps*, and *A. marginale msp4*, respectively.

Corresponding author
Nattawooti Sthitmatee,
nattawooti.s@cmu.ac.th

**Conclusion:** In this study, the analyzed incidence data of cattle hemoparasitic infection in Thailand has provided valuable and basic information for the adaptation of blood-borne parasitic infections control strategies. Moreover, the data obtained from this study would be useful for future effective parasitic disease prevention and surveillance among cattle.

## INTRODUCTION

Bovine hemoparasitic diseases, such as babesiosis, theileriosis, and anaplasmosis, are widely distributed throughout tropical and sub-tropical regions including Thailand. Most of these hemoparasitic diseases are tick-borne diseases and can adversely impact animal health, the livestock industry, and on occasion, human beings. Infections can be deadly to farm animals but are also known to be the cause of fever, anorexia, jaundice, increased abortion rates, and sterility (*Abdullah et al., 2019*). Bovine babesiosis is a serious challenge to the health of farm animals and is caused by a protozoan parasite of the genus *Babesia* found in the erythrocyte. Two species, *Babesia bovis* and *Babesia bigemina*, are known to be extremely prevalent throughout their geographical distribution (*Bock et al., 2004*; *Sawczuk, 2007*), while other species, such as *Babesia divergens*, *Babesia major*, *Babesia jakimovi*, *Babesia ovata*, *Babesia occultans*, and *Babesia mymensingh*, have also been implicated in cattle infections (*Chauvin et al., 2009*; *Sivakumar et al., 2018*).

Bovine theileriosis is a hemoparasitic disease caused by protozoans of the genus *Theileria*. This protozoan is found in the blood and lymphatic systems of infected animals. *Theileria orientalis*, *Theileria annulata*, *Theileria parva*, *Theileria taurotragi*, and *Theileria velifera* are known to be the cause of bovine theileriosis (*Abdullah et al., 2019*; *Olds, Mason & Scoles, 2018*). *T. annulata* and *T. parva* are highly virulent lympho-proliferative parasites that cause tropical theileriosis and East Coast fever, respectively (*Mukhebi, Perry & Kruska, 1992*). *T. orientalis* is a non-lymphoproliferative *Theileria* parasite that is widely distributed throughout Southeast Asia (*Kamau et al., 2011*; *McFadden et al., 2011*).

Bovine anaplasmosis is another tick-borne disease caused by a rickettsia of the *Anaplasmataceae* family. *Anaplasma marginale*, *Anaplasma phagocytophilum*, and *Anaplasma centrale* are important species that are known to infect cattle (*Hornok et al., 2007*; *Kocan, Blouin & Barbet, 2000*). *A. marginale* is the most prevalent tick-borne parasite of cattle worldwide (*Kocan et al., 2010*). Accordingly, there have been many reports of *Babesia* spp., *Theileria* spp., and *Anaplasma* spp. co-infections in cattle (*Altay et al., 2008*; *Bursakov & Kovalchuk, 2019*; *Nyabongo et al., 2021*; *Suarez & Noh, 2011*; *Zhou et al., 2019*).

The occurrence of bovine hemoparasitic infection has been reported in different parts of Thailand (*Altangerel et al., 2011*; *Jirapattharasate et al., 2016*; *Jirapattharasate et al., 2017*; *Simking et al., 2013*; *Sukhumsirichart, Uthaisang-Tanechpongtamb & Chansiri, 1999*). According to these studies, there is an interesting report found that *Babesia* spp. and

*T. orientalis* are endemic in cattle from the western region and that *A. marginale* was the most prevalent pathogen in beef cattle from the north, northeastern, and western of Thailand (*Jirapattharasate et al., 2017*). Hence, this study investigated the parasitic epidemiology of north and central Thailand to improve the general understanding of these infections and to contribute towards effective efforts of strategic control.

## MATERIALS AND METHODS

### Sample and data collection

This study was conducted between June, 2020 and April, 2021. Dairy cattle farms and beef cattle farms with high population densities that are located in six provinces in northern and central Thailand were selected for this study. The sample-collection protocols were reviewed and approved by Animal Care and Use Committee at Faculty of Veterinary Medicine, Chiang Mai University (S26/2563). Farm owner permission letters were approved before samples were collected. The provinces included in this study were Chiang Mai ($n = 143$), Chiang Rai ($n = 87$), Lamphun ($n = 557$), Lampang ($n = 76$), Phayao ($n = 122$), and Nakhon Pathom ($n = 81$) (Fig. 1). A total of 1,066 blood samples were collected from randomly selected farms located in five provinces in northern Thailand and another province in central Thailand. The animals were restrained and blood was collected from their coccygeal or jugular veins and immediately transferred into EDTA-K2 lyophilized vacuum blood collection tubes (BD Vacutainer®, Franklin Lakes, NJ, USA). The blood sample tubes were kept in a cooled box equipped with ice packs during transport to the Faculty of Veterinary Medicine, Chiang Mai University and processed immediately. Data related to the characteristics of animals and farm management were obtained and recorded by the investigators. At each farm, farm owners or farm staff were interviewed with regard to specific individual animal characteristics, namely age, breed, and gender. Farm-based characteristics included location, history of hemoparasitic infection, treatment details, tick control programs, and farm management practices.

### Microscopic analysis

A thin smear of blood was prepared from each blood sample. The blood smears were air-dried, fixed in methanol for 2 min, and stained by 10% Giemsa solution (Merck, Kenilworth, NJ, USA) in phosphate-buffered saline. The smears were examined at 1,000× magnification using an oil-immersion lens (CX31; Olympus, Shinjuku City, Tokyo, Japan). The identification process was carried out to decipher genus and species profiling to the greatest degree possible. A minimum of 1,000 red blood cells were counted and recorded. The percent of parasitemia was determined by counting the number of infected red blood cells (iRBCs) and by then dividing that number by the total number of red blood cells (RBCs): % parasitemia = (iRBCs/RBCs) × 100.

### DNA extraction from blood

Genomic DNA was extracted from all blood samples using a PureLink™ Genomic DNA mini kit (Invitrogen, Thermo Fisher Scientific, Waltham, MA, USA) in accordance with the manufacturer's instructions. The extracted DNA was eluted in 70 μL of the elution

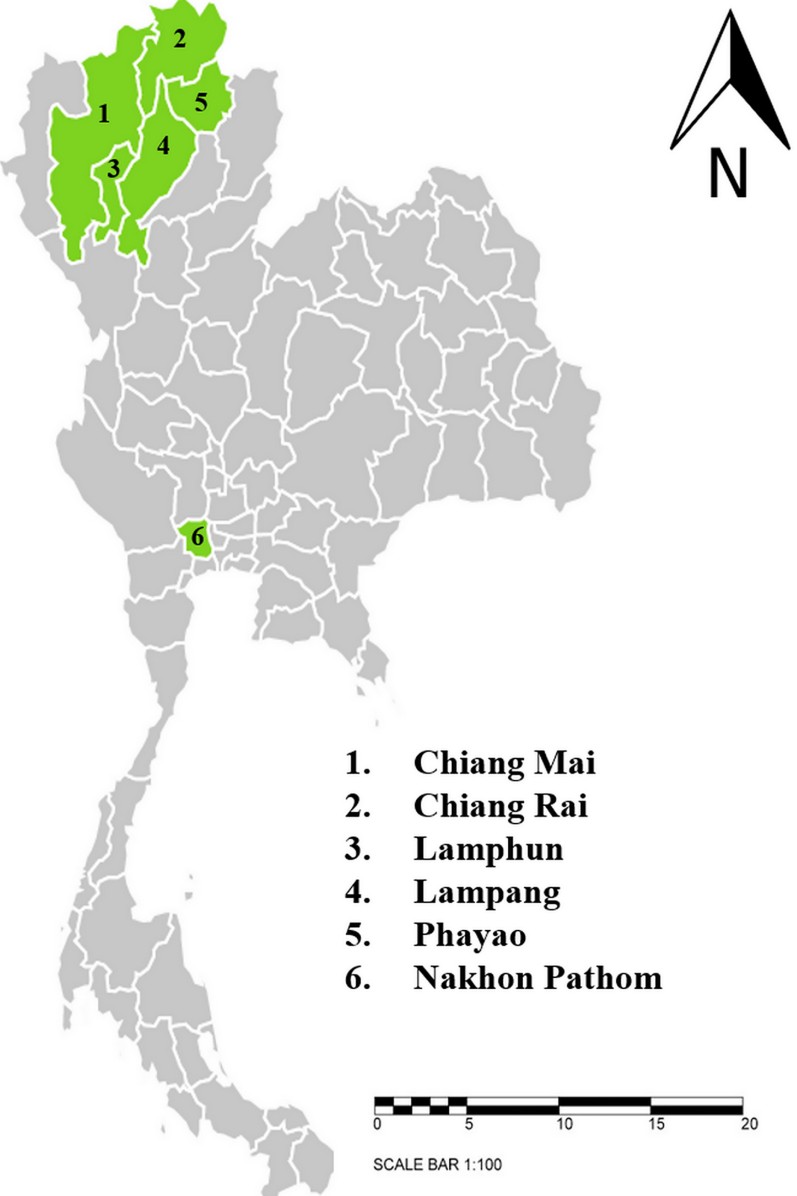

**Figure 1 Sampling areas map in northern and central Thailand.** A total 1,066 blood samples were collected from 6 provinces; 1. Chiang Mai ($n = 143$), 2. Chiang Rai ($n = 87$), 3. Lamphun ($n = 557$), 4. Lampang ($n = 76$), 5. Phayao ($n = 122$) and 6. Nakhon Pathom ($n = 81$). The map using an online infographic tool for map generation (https://create.piktochart.com).

buffer. The quantity and quality of the DNA were determined by an UV/Vis spectrophotometer DU 730 (Beckman Coulter, Brea, CA, USA). The DNA was stored at −20 °C until it was used.

## Packed cell volume determination using microhematocrit method

Blood samples were pipetted into capillary tubes and spun in a high-speed centrifuge. After 5 min of centrifugation, hematocrit results were estimated by calculating the ratio of the

column of packed erythrocytes to the total length of the sample in the capillary tube and then measured with a hematocrit reader card. The reference value was determined to be between 24% and 46% (*Miller et al., 1989*).

## PCR detection of cattle hemoparasites

All samples were analyzed by PCR parallel with microscopic analysis. For the purposes of PCR analysis, *Babesia* spp. and *Theileria* spp. parasites were screened for the presence of genetic differences among *Babesia* 18S rRNA (*Hilpertshauser et al., 2006*) and *Theileria* spp. 18S rRNA (*Cao et al., 2013*). Positive control samples were included for each specific screen. *B. bovis* was determined by *B. bovis* rhoptry-associated protein (*rap-1*) (*Figueroa et al., 1993*), *B. bigemina* was determined by *B. bigemina* apical membrane antigen 1 (*ama-1*) (*Sivakumar et al., 2012*). *T. orientalis* was determined by *T. orientalis* major piroplasm surface protein (*mpsp*) (*Ota et al., 2009*), and *T. annulata* was determined by *T. annulata* 30 kDa major merozoite surface antigen gene (*tams-1*) (*Kirvar et al., 2000*). In addition, the prevalence of *A. marginale* and *A. phagocytophilum* were determined by genetic variations of *A. marginale* major surface protein 4 (*msp4*) (*M'Ghirbi et al., 2016*) and *A. phagocytophilum* major surface protein 2 (*msp2*) (*M'Ghirbi et al., 2016*).

DNA from each sample was PCR amplified using the gene-specific primers for which the sequences is listed in Table 1. *Babesia* spp. and *Theileria* spp. parasites were further screened using nested PCR (nPCR). Positive samples were specifically screened, while *B. bovis* and *B. bigemina* were screened by nPCR. Meanwhile, *T. orientalis, T. annulata, A. marginale*, and *A. phagocytophilum* parasites were screened by primary PCR. The final volume of 30 µL was comprised of 5 µL of template DNA and 25 µL of the reaction mixture with 2X MyTaq HS Red Mix (Meridian Bioscience, Bioline, Memphis, TN, USA) and 10 µM of each sample, which was made up with deionized water to reach the final volume.

The conditions used for *Babesia* spp. and *Theileria* spp. amplification consisted of initial denaturing at 95 °C for 1 min, 35 cycles of a denaturing step at 95 °C for 15 s, an annealing step for *Babesia* spp. at 63 °C for 15 s and *Theileria* spp. at 55 °C for 30 s, an extension step at 72 °C for 15 s, and a final extension step at 72 °C for 30 s. The same concentration of MyTaq HS Red Mix was used for *Theileria* spp. amplification of 5 µL PCR product in the nPCR as has been described above. The nPCR condition included initial denaturation at 95 °C for 1 min and 35 cycles of a denaturing step at 95 °C for 15 s, annealing temperatures of 60 °C for 10 s for *Babesia spp* with an extension step at 72 °C for 10 s, and a final extension step at 72 °C for 30 s. Then, *Babesia* spp. positive samples were identified for *B. bovis* and *B. bigemina*. The same concentration of MyTaq HS Red Mix that was used for amplification consisted of an initial denaturing step at 95 °C for 1 min, 35 cycles of a denaturing step at 95°C for 15 s, an annealing step for *B. bovis* at 55 °C for 15 s and *B. bigemina* at 61 °C for 15 s, an extension step at 72 °C for 15 s, and a final extension step at 72 °C for 30 s. The same conditions used for *B. bovis* and *B. bigemina* amplification of 5 µL PCR product in the nPCR were used for the first PCR of each strain. While, *Theileria* spp. positive samples were identified for *T. orientalis* and *T. annulata*. Identification of *T. orientalis* and *A. phagocytophilum* was performed in a PCR thermal

**Table 1  Forward and reverse primers used for the detection of cattle hemoparasitic infection.**

| Species | Target gene | Oligonucleotide sequence (5′→ 3′) | Size (bp) | Ref. |
|---|---|---|---|---|
| *Babesia spp.* | 18S rRNA | Outer forward: GTTTCTGMCCCATCAGCTTGAC | 1,201–1,248 | *Hilpertshauser et al. (2006)* |
| | | Outer reverse: GCATACTAGGCATTCCTCGTTCAT | | |
| | | Inner forward: GTTTCTGMCCCATCAGCTTGAC | 494–528 | |
| | | Inner reverse: CAACCGTTCCTATTAACCATTAC | | |
| *B. bovis* | rap-1 | Outer forward: CACGAGGAAGGAACTACCGATGTTGA | 365 | *Figueroa et al. (1993)* |
| | | Outer reverse: CCAAGGAGCTTCAACGTACGAGGTCA | | |
| | | Inner forward: TCAACAAGGTACTCTATATGGCTACC | 298 | |
| | | Inner reverse: CTACCGAGCAGAACCTTCTTCACCAT | | |
| *B. bigemina* | ama-1 | Outer forward: TCGGCAGGTGCTCTTACAAAC | 711 | *Sivakumar et al. (2012)* |
| | | Outer reverse: GTTCAGGATACGGCAAACACC | | |
| | | Inner forward: ATTTGTCGCCAGTATCAGCCG | 480 | |
| | | Inner reverse: CAATGTCAACATCCGCAGCTG | | |
| *Theileria spp.* | 18S rRNA | Outer forward: GAAACGGCTACCACATCT | 778 | *Cao et al. (2013)* |
| | | Outer reverse: AGTTTCCCCGTGTTGAGT | | |
| | | Inner forward: TTAAACCTCTTCCAGAGT | 581 | |
| | | Inner reverse: TCAGCCTTGCGACCATAC | | |
| *T. orientalis* | mpsp | forward: CTTTGCCTAGGATACTTCCT | 776 | *Ota et al. (2009)* |
| | | reverse: ACGGCAAGTGGTGAGAACT | | |
| *T. annulata* | tams-1 | forward: ATGCTGCAAATGAGGAT | 785 | *Kirvar et al. (2000)* |
| | | reverse: GGACTGATGAGAAGACGATGAG | | |
| *A. marginale* | msp4 | forward: ATCTTTCGACGGCGCTGTG | 420 | *M'Ghirbi et al. (2016)* |
| | | reverse: ATGTCCTTGTAAGACTCATCAAATAGC | | |
| *A. phagocytophilum* | msp2 | forward: CCAGCGTTTAGCAAGATAAGAG | 334 | *M'Ghirbi et al. (2016)* |
| | | reverse: GCCCAGTAACAACATCATAAGC | | |

cycler consisting of initial denaturation at 94 °C for 3 min, 40 cycles of a second denaturation step at 94 °C for 1 min, an annealing step at 58 °C for 30 s, an extension step at 72 °C for 1 min, and a final extension step at 72 °C for 5 min. Similarly, in terms of the PCR specification of *A. marginale* and *T. annulata*., the initial denaturation step was set at 94 °C for 3 min, 40 cycles of a second denaturation step at 94 °C for 1 min, an annealing step at 60 °C for 30 s, an extension step at 72 °C for 1 min, and a final extension step at 72 °C for 5 min. All PCR products were separated by gel electrophoresis on 1% agarose in 1X TAE buffer and visualized using ethidium bromide under UV transilluminator.

## DNA sequencing and phylogenetic tree analysis

*B. bovis* ($n = 4$), *T. orientalis* ($n = 12$), and *A. marginale* ($n = 5$) positive samples were randomly selected for DNA sequencing. PCR products were purified using PureLink™ quick PCR purification kit (Invitrogen, Thermo Fisher Scientific, Waltham, MA, USA). The purified PCR samples were sent to ATCC Co. Ltd. (Thailand Science Park, Khlong Nueng, Thailand) for identification of species by DNA sequencing. Nucleotide sequences were analyzed using the BLAST tool on the Clustal 2.1 software. The completed sequences were subjected to multiple sequence alignment with sequences previously available in the

**(A)**  **(B)**  **(C)**

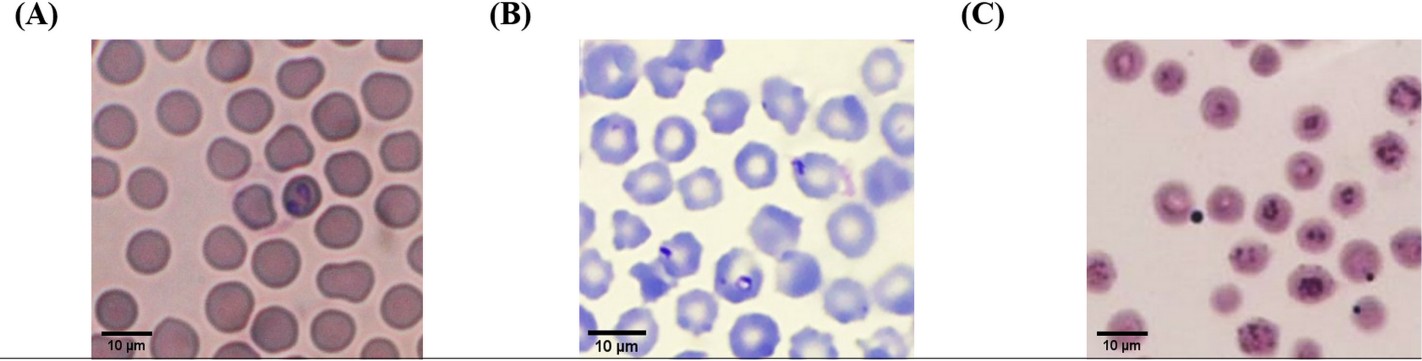

**Figure 2 Typical morphology of cattle hemoparasitic infection in a thin blood smear stained with 10% Giemsa showing multiple infected RBCs.** (A) *Babesia* spp.; (B) *Theileria* spp.; (C) *Anaplasma* spp.

GenBank database. Phylogenetic trees in this study were analyzed using MEGA X program. Initial trees for the heuristic search were obtained automatically by applying Neighbor-Joining method and BioNJ algorithms to a matrix of pairwise distances estimated using the Tamura-Nei model, and then selecting the topology with superior log likelihood value. The tree is drawn to scale, with branch lengths measured in the number of substitutions per site. *Rap-1* gene sequences of *B. bovis* (*n* = 4), *mpsp* gene sequences of *T. orientalis* (*n* = 12), and *msp4* gene sequences of *A. marginale* (*n* = 5), while those reported from other regions were used to construct a subsequent phylogenetic tree. Bootstrap test with 2,000 replications was established as the confidence of the branching pattern of the trees. Finally, the phylogenetic relationship among the isolates identified in this study and those isolated from different countries were illustrated.

### Statistical analysis

Statistical analysis of data categorized as positive or negative for *B. bovis, T. orientalis*, and *A. marginale* was accomplished based on PCR results and the packed cell volume. Variables were performed using the chi-square test. A *P*-value of <0.05 was considered to be statistically significant using GraphPad Prism version 8.4.

## RESULTS

### Microscopic examination of cattle hemoparasitic infections

According to light microscopic examinations, variable cattle hemoparasites, such as *Babesia* spp., *Theileria* spp., and *Anaplasma* spp., were detected in Giemsa-stained blood smears (Fig. 2). Paired-pyriform parasites within the erythrocyte were observed explicitly as a characteristic of *Babesia* spp. The pyriform shape of the *Theileria* parasites was clearly detected. A small spot located on the edge or center of the red blood cell was confirmed as a characteristic form of *Anaplasma* spp.

### Hematological examination

Accordingly, 965 blood samples of total 1,066 samples were examined. The results indicated that 161 hemoparasite infected samples exhibited positive indications of anemia, as is shown in Table 2.

**Table 2 Packed cell volume determination by the microhematocrit method.**

| Species | Packed cell volume | Hemoparasitic infection | | Chi square | P-value |
|---|---|---|---|---|---|
| | | positive | negative | | |
| Babesia spp. | <24% | 3 | 127 | 0.01221 | 0.9120 |
| | 24–46% | 18 | 817 | | |
| Theileria spp. | <24% | 103 | 27 | 3.187 | 0.0742 |
| | 24–46% | 599 | 236 | | |
| Anaplasma spp. | <24% | 55 | 75 | 1.409 | 0.2352 |
| | 24–46% | 308 | 527 | | |

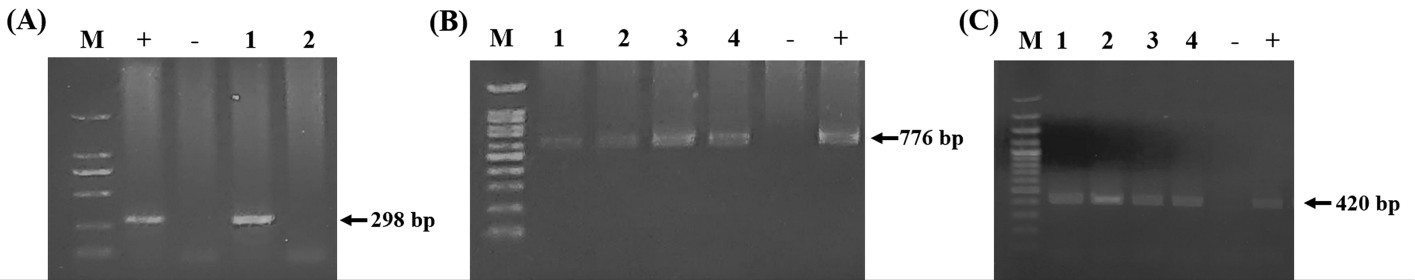

**Figure 3 PCR detection of cattle hemoparasite infection.** (A) *B. bovis* (298 bp) (B) *T. orientalis* (776 bp) (C) *A. marginale* (420 bp). The molecular size standard is a 100 bp ladder, the number indicated that tested samples, positive and negative control DNA were also indicated.

## Molecular detection and identification of cattle hemoparasites

Using the specific primers, PCR products at 298, 776, and 420 bp were determined to represent *B. bovis, T. orientalis*, and *A. marginale*, respectively (Fig. 3). Meanwhile, *B. bigemina, T. annulata*, and *A. phagocytophilum* were not detected. The PCR results indicated that 1.22% with 95% CI [0.56–1.88] (13/1,066) of the blood samples were positive for *B. bovis, T. orientalis*, and *A. marginale* at 36.50% with 95% CI [33.60–39.38] (389/1,066) and 34.15% with 95% CI [31.30–36.99] (364/1,066), respectively, as is shown in Table 3. Furthermore, the multiple infections of two or more cattle hemoparasites appeared in 27.30% with 95% CI [ 24.62–29.97] (291/1,066) of the total blood samples (Fig. 4). By the presence of multiple infections, *T. orientalis* at 99.66% with 95% CI [98.98–100] (290/291) was found to be the most frequent hemoparasite.

## DNA sequencing and phylogenetic analysis of cattle hemoparasites

The molecular characterizations of cattle hemoparasites were analyzed with respective gene targets of *B. bovis rap -1, T. orientalis mpsp*, and *A. marginale msp4*. The identity of the *B. bovis* specificity among isolates in this study ranged between 98.57–100%. Four isolates obtained from Lamphun (OK490920), Lampang (OK490921 and KO490922), and Nakhon Pathom (OK490919) provinces with a product size of 298 bp shared a degree of similarity with the isolates obtained from China (KT312809.1) and the Philippines (JX860283.1) (Fig. 5).

**Table 3 Summary of PCR screening results for *B. bovis, T. orientalis*, and *A. marginale* single infections in cattle from northern and central Thailand.**

| Province | No. of cattle | B. bovis | | T. orientalis | | A. marginale | |
|---|---|---|---|---|---|---|---|
| | | Positive | % | Positive | % | Positive | % |
| Chiang Mai | 143 | 0 | 0 | 71 | 49.65 | 5 | 3.50 |
| Chiang Rai | 87 | 1 | 1.15 | 2 | 2.30 | 2 | 2.30 |
| Lamphun | 557 | 7 | 1.26 | 200 | 35.90 | 187 | 33.57 |
| Lampang | 76 | 4 | 5.26 | 16 | 21.05 | 65 | 85.53 |
| Phayao | 122 | 0 | 0 | 92 | 75.41 | 92 | 75.41 |
| Nakhon Pathom | 81 | 1 | 1.23 | 8 | 9.87 | 13 | 16.05 |
| **Total** | **1,066** | **13** | **1.22** | **389** | **36.50** | **364** | **34.15** |

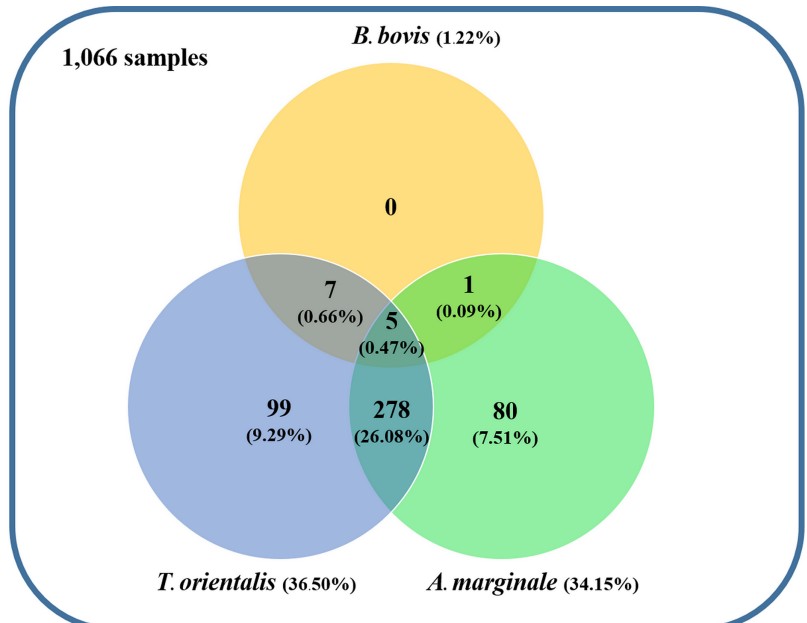

**Figure 4 Venn diagram summarizing the species specificity and infection rate of cattle hemoparasite infections in northern and central Thailand.**

Further characterizations of *T. orientalis* and *A. marginale* were identified using phylogenetic analysis with PCR assay. The product size of 776 bp in this study was placed in four isolated genotypes by the percent identities ranging between 83.96–100% (Fig. 6). The phylogenetics of six isolates obtained from Lamphun (OK490929), Lampang (OK490926 and OK490928), Phayao (OK490927 and OK490930), and Nakhon Pathom (OK490931) provinces were related to *T. orientalis* type 5 and shared similarity with isolates obtained from several areas on the Asian continent. There were two isolates obtained from Phayao (OK490923) and Nakhon Pathom (OK490924) provinces that were related to the *T. orientalis* type 3 and shared similarity with isolates obtained from Sri-Lanka (AB701465) and Vietnam (AB560821). Moreover, another sequence result

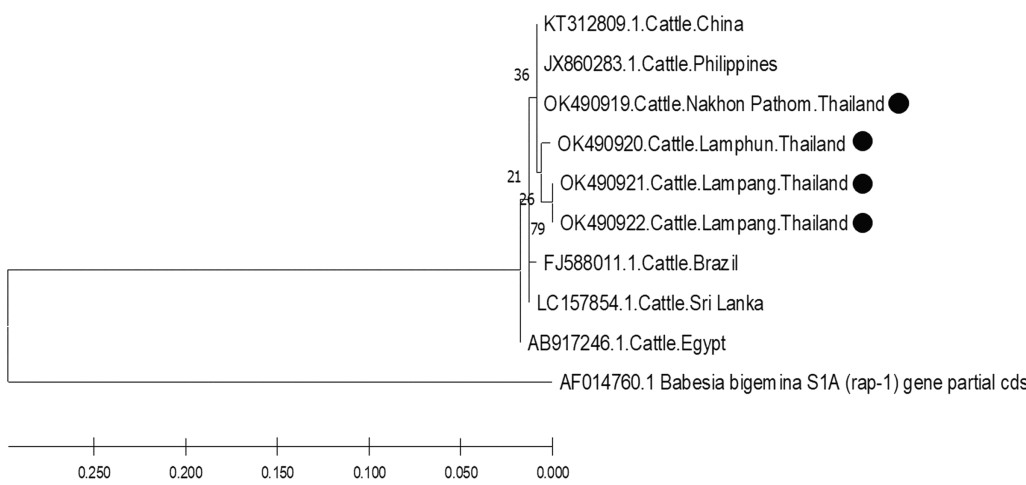

**Figure 5 Phylogenetic relationships based on *rap-1* sequence of *B. bovis*, in accordance with the PCR amplified sequence.** The evolutionary history was inferred *via* the neighbor-joining method. The percentage of replicate trees wherein the associated taxa clustered together in the bootstrap test (2,000 replicates) are shown next to the branches. Evolutionary analyzes were conducted using MEGAX (•*B. bovis* positive in this study) was used *B. bigemina* S1A (*rap-1*) gene partial cds as outgroup.

obtained from Lamphun (OK490925) province revealed the presence of *T. orientalis* type 4, which has been reported to be present in China (MH539832), Myanmar (AB871316), and Thailand (AB562561). *T. orientalis* type 7 was found lastly in this study, while three isolates from Chiang Mai (OK490933), Chiang Rai (OK490934), and Nakhon Pathom (OK490932) provinces exhibited similarity with the databases established from Japan (AB218430), China (MH539826), Indonesia (AF102500), and Vietnam (AB560823).

Finally, the PCR product size of 420 bp confirmed the presence of *A. marginale*. The phylogenetic findings of five isolates obtained from Lamphun (OK506074, OK506075, and OK506077), Chiang Rai (OK506076) and Nakhon Pathom (OK506073) provinces revealed 97.60–100% of sequence similarity when compared to isolates obtained from Brazil (JN022561), Colombia (MF771065), and South Africa (KF758944 and MT173811) (Fig. 7).

## DISCUSSION

Thailand is known to be an endemic area for various bovine tick-borne pathogens, which can affect the health of farm animals and cause significant economic losses to the livestock industry. Incidence studies involving cattle hemoparasitic infections could provide valuable basic information to contribute towards effective efforts of strategic control. In this study, microscopic analysis, which is the worldwide standard protocol, was performed for primary detection of these parasites. However, parasitemia was very low, while morphological differentiations of various *Theileria* spp. and *Babesia* spp. ring forms were inconclusive. Consequently, molecular tools are needed to verify complementary diagnostic information with a high degree of specificity and sensitivity.

The presence of *Babesia* spp. and *Theileria* spp. was first determined using 18S primers. Afterward, species of each genus, namely *B. bovis, T. orientalis,* and *A. marginale,* were

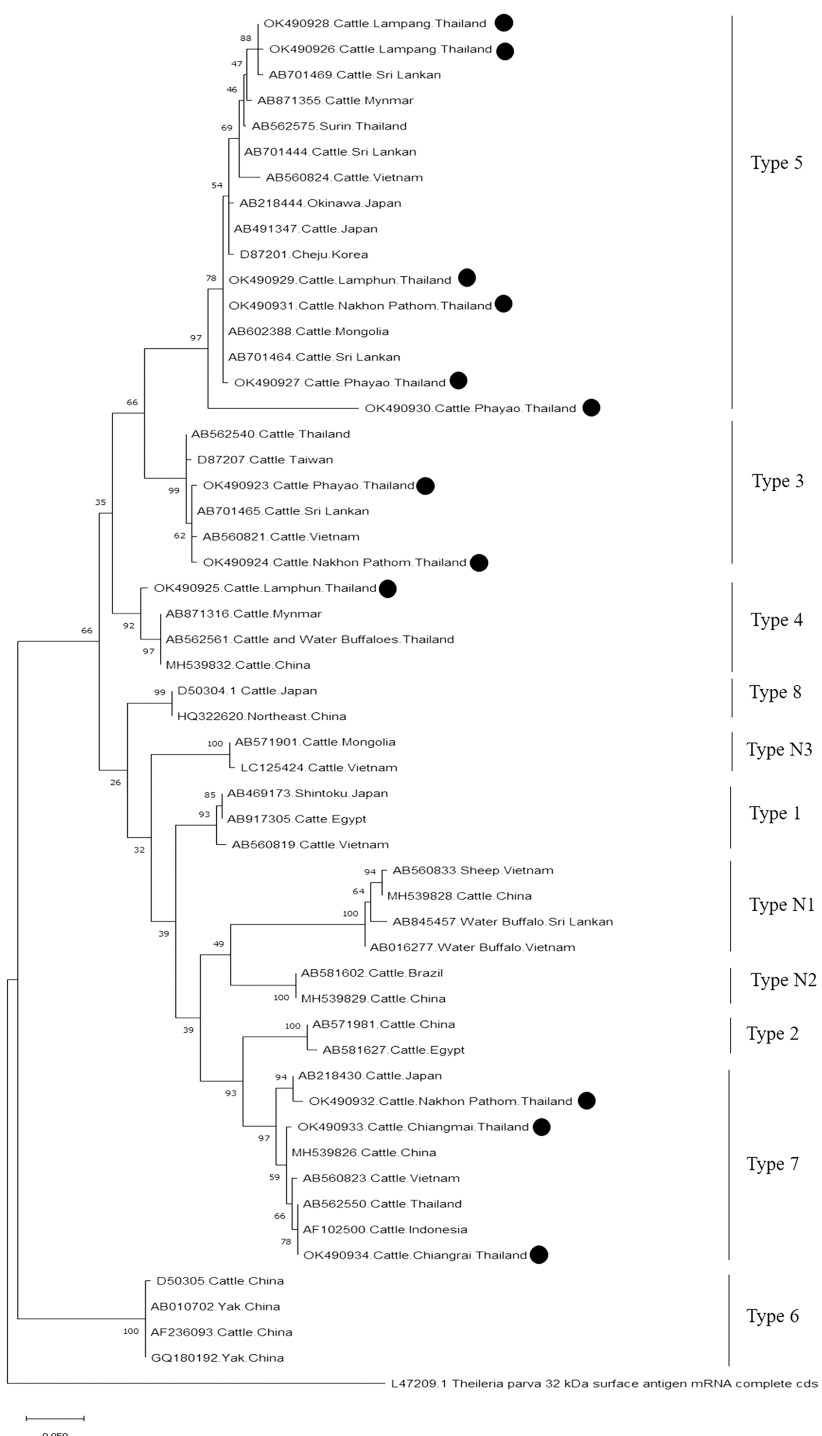

**Figure 6 Phylogenetic relationships based on *mpsp* sequence of *T. orientalis*, in accordance with the PCR amplified sequence.** The evolutionary history was inferred *via* the neighbor-joining method. The percentage of replicate trees wherein the associated taxa clustered together in the bootstrap test (2,000 replicates) are shown next to the branches. Evolutionary analyzes were conducted using MEGAX (•*T. orientalis* positive in this study) was used *T. parva* 32 kDa surface antigen mRNA complete cds as outgroup.

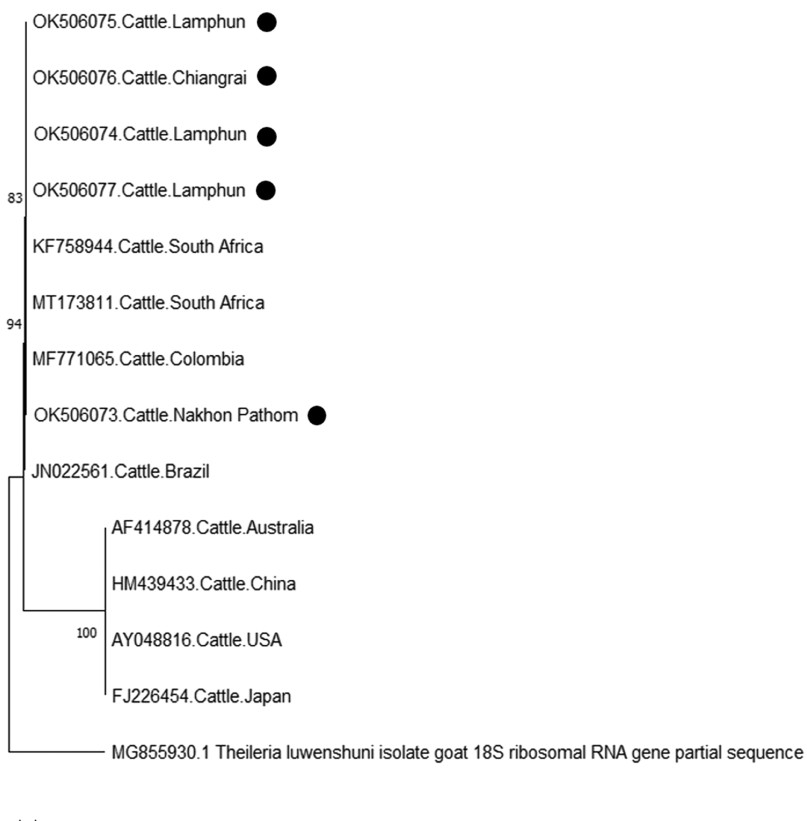

**Figure 7 Phylogenetic relationships based on *msp4* sequence of *A. marginale*, in accordance with the PCR amplified sequence.** The evolutionary history was inferred *via* the neighbor-joining method. The percentage of replicate trees wherein the associated taxa clustered together in the bootstrap test (2,000 replicates) are shown next to the branches. Evolutionary analyzes were conducted using MEGAX (•*A. marginale* positive in this study) was used *T. luwenshuni* isolate goat 18S ribosomal RNA gene partial sequence as outgroup.

identified by specific screening, as described in the methods section. In the present study, the positive rates of those hemoparasites were variable for the different sampling sites in Thailand. Importantly, the sampling period, tick control program, and farming management practices were also related to the positive results. Although the hematocrit results detected anemia in the hemoparasitic infected samples, no significant differences were observed (Table 2). It is important to recognize that hemoparasitic infected animals can go undetected as they may be tested in the parasitic incubation period, which could result in low parasitemia, the absence of clinical signs, and normal hematocrit results. Consequently, the reported percent of hematocrit may possibly be related to the dehydration of the animal.

Previous epidemiological studies conducted in Thailand have helped to identify and manage the relevant burden and risk factors associated with incidences of tick-borne diseases (*Jirapattharasate et al., 2016*; *Jirapattharasate et al., 2017*). In this study, PCR was used as a specific tool for *B. bovis*, *T. orientalis*, and *A. marginale* detection because it has been reported to be a highly specific and sensitive method (*Altay et al., 2008*). Overall, the

sampled cattle had at least one incidence of hemoparasite (*B. bovis* (1.22%), *T. orientalis* (36.50%), and *A. marginale* (34.15%)) infection. These detections were not significantly different from those of previous studies conducted in Thailand, wherein the prevalence of the above parasites ranged from 0.8% to 31.0% (*Altangerel et al., 2011*; *Jirapattharasate et al., 2017*; *Sarataphan et al., 2003*). Furthermore, there have been some reports of a female tick vector, *Rhipicephalus (Boophilus) microplus*, which has exhibited a higher frequency of infection with *B. bigemina* than *B. bovis*. Hence, the chance of *B. bigemina* transmission by the tick vector is higher than *B. bovis* (*Oliveira-Sequeira et al., 2005*; *Oliveira et al., 2008*). Although, previous studies reported a higher occurrence of *B. bigemina* than *B. bovis* (*Jirapattharasate et al., 2016*; *Jirapattharasate et al., 2017*), *B. bigemina* went undetected in this study. This might be due to the increased drug resistant prowess of *B. bovis* when compared to *B. bigemina*. The sequence analysis of the *B. bovis* apical membrane antigen 1 (BbAMA-1) obtained from Thai cattle has exhibited a low level of polymorphism among global isolates, while some epitopes were found to infrequently be polymorphic due to amino acid mutations (*Rittipornlertrak et al., 2017*). This problem is indicative of the challenges associated with this vaccine candidate and the process of novel antibabesial drug development. According to the interviews conducted with farmers in this study, the tick control program was especially noteworthy. The cattle at most sampling farms were treated with diminazene aceturate and ivermectin in order to prevent hemoparasitic infection and to avoid establishing a parasite vector. Even though this practice might increase the drug resistance index (*Chaparro-Gutiérrez, Villar & Schaeffer, 2020*; *Chitanga et al., 2011*; *Tuvshintulga et al., 2019*), it is currently a widely-used and pervasive tick control program and farming management practice in Thailand. From the hemoparasite detection results, diminazene aceturate and ivermectin seemed capable of preventing some incidences of cattle hemoparasites. We found minimal *Babesia* spp. infections at all sampling farms, whereas *Theileria* spp. and *Anaplasma* spp. infections remained high. Notably, *T. orientalis* was recognized as the most frequently identified hemoparasite with multiple infections (99.66%). Detection of these hemoparasite is evidence of the need to develop a combined vaccine or drug for the treatment of multi-hemoparasitic infection.

Based on DNA sequencing and the phylogenetic tree findings, *B. bovis rap-1* was highly conserved amongst the cattle samples in the current study and exhibited high correlation with other previously reported geographic isolates. These results confirm that the *rap-1* gene is a useable target for the detection of hemoparasites from different geographic areas (*Figueroa et al., 1993*). The phylogenetic tree of *B. bovis* isolates in these three provinces indicated that the *rap-1* gene is relatively conserved. It appears that *B. bovis* isolates obtained from northern and central Thailand were of the same strain with other geographic areas. Although, Nakhon Pathom is located in central Thailand, the *rap-1* gene isolate from this location was identified with isolates collected from northern provinces, namely Lamphun and Lampang, as these areas are located on the same continent as China and the Philippines. Hence, an effective approach for disease tracking will be beneficial as a control strategy for bovine babesiosis in these locations.

Moreover, phylogenetic analysis in this study also revealed that *T. orientalis mpsp* gene sequences were classified into four clades (type 3, type 4, type 5, and type 7), which was similar to the findings of a previous report (*Altangerel et al., 2011*; *Jirapattharasate et al., 2017*). This result confirmed that the *mpsp* gene is a highly polymorphic gene that exhibited wide range of diversity among the different filed isolates (*Sivakumar et al., 2014*). In this study, we also found that cattle from every sampling farm were positive for *Theileria* spp. infection. It could then be inferred that *Theileria* spp. infection is commonly found in these areas. Therefore, good farm management practices and routine tick control campaigns (*L'Hostis & Seegers, 2002*) would help to reduce the prevalence of bovine Theileriosis and other tick-borne parasitic diseases in northern and central Thailand.

According to existing genetic diversity, the nucleotide sequence levels of *A. marginale* were based on the *msp4* gene. Sequences of the *msp4* gene obtained in this study were conserved and aligned with those of previous reports (*Junsiri et al., 2020*). Phylogenetic analysis revealed that all the *msp4* sequences were clustered with sequences obtained from Brazil, Columbia, Portugal, and South Africa. Previous reports on animal movement also suggest that the genetic diversity of *A. marginale* in this study correlated to the incidences of *A. marginale* infection in various other countries (*Jirapattharasate et al., 2017*). Therefore, restricting animal transportation may help to effectively control the genetic diversity of *A. marginale* and other hemoparasites.

## CONCLUSIONS

The distribution of bovine hemoparasites across a wide geographical area of northern and central Thailand has revealed that *T. orientalis* is an endemic hemoparasite among Thai cattle. However, *B. bovis* detection rates decreased from those of previous reports. While *A. marginale* is a highly prevalent pathogen in cattle from the north and central regions of Thailand, these findings can improve the general understanding of the epidemiology of hemoparasites in Thailand and can contribute to the design of effective parasite control strategies in the future.

## ACKNOWLEDGEMENTS

We would like to thank the Faculty of Veterinary Medicine, Chiang Mai University for providing the necessary laboratory facilities. In addition, we thank all owners and members of staff of the farms participating in this study for their kind cooperation.

### Funding

This project was funded by the National Research Council of Thailand (NRCT), grant no. NRCT5-RSA63004. The funders had no role in study design, data collection and analysis, decision to publish, or preparation of the manuscript. The funders had no role in study design, data collection and analysis, decision to publish, or preparation of the manuscript.

## Grant Disclosures

The following grant information was disclosed by the authors:
National Research Council of Thailand (NRCT) Grant No: NRCT5-RSA63004.

## Competing Interests

The authors declare that they have no competing interests.

## Author Contributions

- Pongpisid Koonyosying conceived and designed the experiments, performed the experiments, analyzed the data, prepared figures and/or tables, authored or reviewed drafts of the article, and approved the final draft.
- Amarin Rittipornlertrak performed the experiments, prepared figures and/or tables, and approved the final draft.
- Paweena Chomjit performed the experiments, prepared figures and/or tables, and approved the final draft.
- Kanokwan Sangkakam performed the experiments, prepared figures and/or tables, and approved the final draft.
- Anucha Muenthaisong performed the experiments, prepared figures and/or tables, and approved the final draft.
- Boondarika Nambooppha performed the experiments, prepared figures and/or tables, and approved the final draft.
- Wanwisa Srisawat performed the experiments, prepared figures and/or tables, and approved the final draft.
- Nisachon Apinda performed the experiments, prepared figures and/or tables, and approved the final draft.
- Tawatchai Singhla conceived and designed the experiments, analyzed the data, prepared figures and/or tables, authored or reviewed drafts of the article, and approved the final draft.
- Nattawooti Sthitmatee conceived and designed the experiments, analyzed the data, prepared figures and/or tables, authored or reviewed drafts of the article, and approved the final draft.

## Animal Ethics

The following information was supplied relating to ethical approvals (*i.e.*, approving body and any reference numbers):

All animal protocols in this study were approved and supervised by the Animal Care and Use Committee (FVM-ACUC), Faculty of Veterinary Medicine, Chiang Mai University (Project no. R000028479).

## DNA Deposition

The following information was supplied regarding the deposition of DNA sequences:

The sequences described here are accessible *via* GenBank accession numbers OK490919 to OK490934 and OK506073 to OK506077.

## Data Availability

The raw measurements are available in the Supplemental Files.

## Supplemental Information

Supplemental information for this article can be found online at http://dx.doi.org/10.7717/peerj.13835#supplemental-information.

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
