# Peer review of "Incidence of hemoparasitic infections in cattle from central and northern Thailand"

_PeerJ, doi:10.7717/peerj.13835_

## Round 0.1 · original submission · Minor Revisions

Please see the file attached for my comments.

Reviewer 1 ·

Basic reporting

- English in the whole manuscript needs to be thoroughly edited.
- The scientific name should be written (italicized) in full when first cited in the abstract and in the introduction, then is abbreviated according to standard format.
Need correction in the whole manuscript
- Unify citation style, when citing multiple works (year of publication (the earliest first) or alphabetical order).
- Introduction:
# Line 57-60: add a reference.
# (Abdullah et al. 2019) didn’t investigate the effect of hemoparasites on animals health and the production of milk and meat.

Experimental design

- Line 114: A thin smear of blood was collected?
- Line 135: a graphic reading pad? Do you mean a hematocrit reader card?
- Some references in Table 1 are not related to the listed oligonucleotide sequences.
- Unify the abbreviation for “microliter”. See line 127, 155, 163 and172.
- Line 159-183: Add references when applicable.

Validity of the findings

- Line 175-179: need a reference.
- Line 228: If you calculate the multiple infection among each hemoparasite you will find that 100% (13/13), i.e., for all cases with B. bovis; 78.02% (9284/364) among A. marginale; 74.55% (290/389) among T. orientalis.
- Line 310: Detection of this hemoparasite….. which one?

Additional comments

References:
- Order the reference list entries alphabetically.
- Unify the style of the title according to the journal instructions.
- Unify the style of the Journal name according to the journal instructions
- Ref. 3: Article in Polish.
- Italicize all scientific names.

Reviewer 2 ·

Basic reporting

This article is generally very clear and well written.

Experimental design

I only have minor comments related to experimental design:

1) The introduction discusses the global patterns of hemo-parasitic infections in cattle. It could be improved by refining the introduction to include specific information on what is known in Thailand. For instance, elaborate on the findings stated in lines 85-87.

2) The methods (lines 117-119) could be improved by elaborating on the specific morphology used to identify the parasites, to save on word count the authors could also improve these sections by including texts used to identify these parasites based on morphology.

3) The article would benefit by elaboration on how the phylogenetic trees were constructed. There are many methods that can be used to create phylogenetic trees (e.g., Neighbor-Joining, Bayesian inference, etc). The reproducibility of the methods could be improved by better outlining the methods in lines 193-198.

4) The statistical analyses (lines 200-204) are quite vague. Please specify the exact variable compared between the two groups (i.e, the packed cell volume)?

5) Lines 220-228: The population sample is really a subpopulation of the real population. Please specify standard errors around the prevalences using the sample size.

6) Would it benefit our epidemiological understanding of hemoparasite infections in Thailand by breaking down infection rates by region? I am curious if there are any meaningful spatial patterns in infection patterns.

7) The discussion refers to the proportion positive samples as incidence rate. These samples were not collected over time and hemoparasites can form chronic infections, therefore, it seems more accurate to refer to them as prevalence.

Validity of the findings

no comment

---

## Round 0.2 · Minor Revisions

Thank you for addressing to the changes suggested by the reviewers.
Note that one of the reviewers still has additional comments.

Finally, please attend to the final editorial requested changes.

Reviewer 1 ·

Basic reporting

The scientific names still need correction in Lines 72-88. Write the full genus name in the first citation (e.g. …Two species, Babesia bovis and B. bigemina, are known to be extremely ….., such as B. divergens, B. major, B. jakimovi, B. ovata, .....).

Experimental design

No comments.

Validity of the findings

No comments.

Additional comments

- Line 222 and 224 correct to spp.
- The style of the Journals names still not unified (see Ref. 1, 9, 12, 15, 20, 22, 23, 29, 31).
- Incorrect or incomplete reference details (copied from ResearchGate??): Ref. 6, 34, 38.
- Ref. 9, “trypanosomes” is not a scientific name.
- Italicize scientific name in Ref.23.
- Ref. 32: add, (the article language is Polish and [English Abstract] was used).

---

## Round 0.3 · Minor Revisions

Dear Dr. Sthimatee,

Thank you for attending to the corrections promptly. However, a misunderstanding regarding use of species names still lingers. Pls amend the text such that the first time that a new species is mentioned, its name (including genus) is fully shown. For example, if you first mentioned Babesia bovis, from then on the species will be referred to as B. bovis. However, other other species in the genus Babesia will still be fully spelled out the first time they show up (eg, Babesia bigemina, which after its first time mentioned will be come B. bigemina). Hope that this is clear now. Please amend accordingly.

Thank you!

---

## Round 0.4 · accepted · Accept

Thank you for your prompt corrections!